# Spatial and Functional Roles of Syndecans in Skin Wound Healing

**DOI:** 10.3390/ijms262110571

**Published:** 2025-10-30

**Authors:** Eunhye Park, Han-gyeol Kim, Yowon Un, Eok-Soo Oh

**Affiliations:** Department of Life Sciences, Ewha Womans University, 52, Ewhayeodae-Gil, Seodaemoon-Gu, Seoul 03760, Republic of Korea; dmsgp970327@gmail.com (E.P.); quips1818@ewhain.net (H.-g.K.);

**Keywords:** extracellular matrix, homeostasis, keratinocytes, syndecan, wound healing

## Abstract

Wound healing is a complex, multi-phase process involving hemostasis, inflammation, proliferation, and tissue remodeling. Syndecans (SDCs), a family of transmembrane heparan sulfate proteoglycans, serve as co-receptors for growth factors, cytokines, and ECM components, playing critical roles in cell adhesion, migration, proliferation, and angiogenesis. Among them, SDC-1 and SDC-4 are key regulators of skin wound healing. Due to their distinct spatial and temporal expression across various cell types—such as epithelial cells, fibroblasts, and immune cells—SDCs are well-positioned to coordinate regenerative responses. This review focuses on the spatial regulation of SDCs during skin wound healing, highlighting their roles in epidermal and dermal repair, modulation of intracellular signaling, and remodeling of the wound microenvironment. Overall, SDCs are emerging as central modulators of skin wound healing, with promising implications for regenerative medicine in the skin and beyond.

## 1. Introduction

SDCs (SDCs) are a conserved family of transmembrane heparan sulfate proteoglycans (SDC-1 to SDC-4) that play pivotal roles in mediating cell–cell and cell–matrix interactions [1]. Their extracellular domains are mainly attached to heparan sulfate, enabling them to bind a broad spectrum of ligands, including growth factors (e.g., FGF2 and EGF), extracellular matrix (ECM) proteins (e.g., fibronectin and collagen), and enzymes. This capacity allows SDCs to sense and respond to dynamic changes in the extracellular environment. Acting as a cell surface receptor, they modulate critical signaling pathways governing cellular behaviors such as proliferation, migration, and differentiation [2,3]. Structurally, the transmembrane domain of SDCs anchors them to the plasma membrane and facilitates dimerization, which is essential for signal transduction. Their cytoplasmic domain contains conserved and variable regions that interact with adaptor proteins, such as syntenin and ezrin, linking SDCs to actin remodeling and intracellular signaling cascades. These multifaceted roles enable SDCs to regulate cell shape, motility, and proliferation, contributing to tissue homeostasis and pathological processes, including inflammation, wound healing, and cancer. Each SDC isoform exhibits tissue-specific expression patterns, supporting its specialized functions across different organ systems. SDC-1 is predominantly expressed in epithelial tissues, including the skin, gut, and respiratory tract. SDC-2 is mainly found in fibroblasts and endothelial cells, SDC-3 is enriched in neurons, and SDC-4 is broadly expressed in fibroblasts, endothelial cells, and smooth muscle cells [4].

In the skin, SDCs help maintain structural integrity and cellular function. SDC-1 and SDC-4 are key regulators of keratinocyte differentiation and skin homeostasis. SDC-1 is dynamically expressed during differentiation—upregulated in early to mid-stages and reduced in mature cells—indicating a role in barrier formation [5]. SDC-4 is mainly found in basal and spinous keratinocytes [6], supporting cell–matrix interactions and cytoskeletal organization during early differentiation and repair. In the dermis, SDCs promote fibroblast adhesion, migration, and response to mechanical cues, essential for tissue remodeling. SDC-1 and SDC-4 are expressed in vascular endothelial cells, where they help maintain vascular integrity and barrier function. SDC-1, in particular, interacts with fibrinogen to stabilize it at the cell surface, thereby reducing endothelial permeability [7,8]. In addition, their heparan sulfate chains bind antithrombin III, supporting anticoagulant activity and contributing to vascular homeostasis [9,10]. In the dermis, SDCs promote fibroblast adhesion to ECM components such as fibronectin, collagen, and laminin, and regulate fibroblast migration and mechanosensing, thereby contributing to tissue remodeling and wound repair.

Wound healing is a highly regulated process consisting of four overlapping phases: hemostasis, inflammation, proliferation, and remodeling. Each phase involves complex interactions among epithelial and stromal cells—including keratinocytes, fibroblasts, immune cells, and endothelial cells—as well as ECM components, including remodeling enzymes [11,12]. As both ECM components and cell surface receptors, SDCs are uniquely positioned to regulate key aspects of wound healing. They contribute to immune cell recruitment, fibroblast activation, ECM deposition, angiogenesis, and re-epithelialization by integrating mechanical and biochemical signals from the extracellular environment. Recent studies have further highlighted the therapeutic potential of targeting SDCs in wound repair. For instance, wound-homing peptides such as CAR have been shown to enhance SDC-4 activity by targeting angiogenic vessels [13], underscoring the significance of SDCs in tissue regeneration. Although the role of SDC-4 as an integrin co-receptor in wound healing has been reviewed by Perez et al. [14], this review focuses more broadly on the roles of SDC family members in skin wound healing—particularly their expression and function in epidermal and dermal cells, as well as their involvement in ECM remodeling.

## 2. Skin Wound Healing

In adults, wound healing progresses through four overlapping phases: hemostasis, inflammation, tissue growth, and remodeling. After injury, a clot forms to seal the wound, followed by inflammation, during which neutrophils and macrophages clear debris and pathogens. Macrophages also coordinate granulation tissue formation. During tissue growth, fibroblasts produce ECM components like collagen, endothelial cells drive angiogenesis, and keratinocytes promote re-epithelialization. In the remodeling phase, excess ECM is degraded and tissue structure is restored. These phases involve dynamic regulation of cell adhesion, cytoskeletal changes, and signaling via cytokines and growth factors. Disruption can lead to impaired healing. Given their role in cell signaling, adhesion, and ECM remodeling, SDCs are critical regulators of wound healing, with their expression tightly controlled in space and time (Table 1). The overall stages of skin wound healing and the phase-specific roles of syndecans are illustrated in Figure 1.

### 2.1. Regulation of Hemostasis by SDCs

SDCs are key components of the endothelial glycocalyx, helping maintain vascular integrity and reducing permeability. SDC-4’s heparan sulfate chains bind antithrombin III, contributing to anticoagulation.

Following endothelial injury, various proteolytic enzymes—such as plasmin, thrombin, and matrix metalloproteinases (MMPs)—along with growth factors, including members of the EGF family, become activated. These molecules promote the shedding of SDC ectodomains [15,16]. Shed forms of SDC-1 and SDC-4 have been detected in human dermal wound fluid for up to two days post-injury, with thrombin notably accelerating this shedding process [15]. The shedding of SDC-1 and SDC-4 is regulated through multiple signaling pathways. These include PKC activation via phorbol esters, JNK/SAPK signaling in response to cellular stress, ERK pathway activation through EGF and thrombin receptors, PI3K activation via insulin signaling, and inhibition of protein tyrosine phosphatases by pervanadate. The cleavage is primarily mediated by a TIMP-3-sensitive metalloproteinase [15,16].

Once shed, SDC ectodomains lose their ability to act as co-receptors for growth factors and may instead function as competitive inhibitors, altering downstream signaling dynamics [15]. Interestingly, members of the EGF family not only utilize SDCs for signaling but also induce their shedding [17,18], suggesting a potential negative feedback mechanism. Beyond their shedding-related roles, SDCs contribute to vascular homeostasis and coagulation. SDC-1, for example, binds fibrinogen and stabilizes it at the cell surface, reinforcing endothelial barrier integrity [8]. SDC-4 is implicated in the regulation of platelet activation and aggregation, indicating a broader role in hemostasis beyond the endothelium [19,20]. However, excessive shedding of SDC-1 may promote leukocyte adhesion and increase the risk of thrombosis, linking SDCs to both inflammatory and pro-thrombotic responses [21,22].

### 2.2. Regulation of the Inflammation Phase by SDCs in Skin Wound Healing

During the inflammation phase of skin wound healing, immune cells such as neutrophils and macrophages are rapidly recruited to the injury site to eliminate pathogens and clear necrotic tissue. Neutrophils kill bacteria and secrete pro-inflammatory cytokines like IL-1β and TNF-α to amplify the local immune response. Shortly after, macrophages infiltrate the wound and polarize into pro-inflammatory M1 or anti-inflammatory M2 phenotypes. M1 macrophages sustain inflammation and pathogen clearance early on by producing cytokines such as TNF-α and IL-1β. As healing progresses, M2 macrophages predominate, releasing anti-inflammatory cytokines (IL-10 and TGF-β) that resolve inflammation and promote tissue repair, enabling transition to the proliferation phase.

Notably, SDCs play key modulatory roles during this phase. SDC-1 regulates leukocyte recruitment and activity. In SDC-1-deficient mice, there is excessive and prolonged leukocyte infiltration, increased edema, and elevated expression of pro-inflammatory cytokines (TNF-α and IL-6), chemokines (CCL5 and CCL3), and adhesion molecules (ICAM-1), indicating that SDC-1 acts as a negative regulator restraining excessive inflammation early after injury [23]. On the contrary, SDC-4 modulates eosinophil migration via interaction with antithrombin III (AT-III). While AT-III alone promotes eosinophil movement, it suppresses eotaxin-induced chemotaxis through an SDC4-dependent mechanism. This effect is abolished by enzymatic digestion of SDC-4 or anti-SDC-4 antibody, suggesting that SDC-4 serves as a functional receptor for AT-III. SDC-4 is also known to mediate eosinophil motility through signaling pathways involving protein kinase C, PI3-kinase, and phosphodiesterase [24].

Mechanistically, SDC-1 contributes to the spatial organization of chemokine gradients essential for directed leukocyte migration. Their heparan sulfate chains bind chemokines like IL-8, forming haptotactic gradients that guide neutrophils from blood vessels to the wound. SDC-1-bound IL-8 also enhances neutrophil–endothelial interactions and transmigration [25,26,27].

SDCs’ function is further regulated by extracellular domain shedding. During the inflammation phase of skin wound healing, SDC-1 expression is initially upregulated in endothelial cells and keratinocytes at the wound site and subsequently undergoes proteolytic shedding [28]. Proteases such as plasmin, thrombin, and EGF family members cleave SDCs from cell surfaces, releasing soluble ectodomains (shed SDC-1 and SDC-4). Shed SDC-1 sequesters IL-8 and disrupts its gradient. This reduces directional cues for neutrophil migration and impairs transendothelial migration [15,28]. In human atopic dermatitis, elevated granzyme K (GzmK) cleaves SDC-1, disrupting the glycocalyx and increasing VEGF secretion by keratinocytes. This promotes microvascular damage and microhemorrhage, exacerbating inflammation severity without significantly increasing cytokine production or impairing the epidermal barrier [29]. Therefore, the balanced expression and shedding of SDCs critically modulate leukocyte recruitment, chemokine gradients, and inflammatory resolution during skin wound healing.

### 2.3. Regulation of the Proliferation Phase by SDCs

The proliferation phase of wound healing involves fibroblast activation, angiogenesis, re-epithelialization, and ECM deposition—processes largely driven by growth factors and cell–ECM interactions [30]. M2 macrophages promote granulation tissue formation by secreting key factors: VEGF (angiogenesis), TGF-β (fibroblast proliferation and collagen synthesis), and PDGF (fibroblast recruitment and ECM production) [31]. Fibroblasts migrate to the wound site and produce ECM components such as collagen and fibronectin in response to PDGF and TGF-β [32,33]. Meanwhile, keratinocytes, stimulated by EGF and KGF, proliferate and migrate to restore the skin barrier [34,35,36]. ECM remodeling is dynamically regulated by MMPs and their inhibitors (TIMPs), maintaining a balance between synthesis and degradation [37,38]. Together, these events rebuild tissue structure and prepare it for the remodeling phase. Among key regulators of this phase are SDC-1 and SDC-4, which modulate multiple aspects of cellular activity during repair.

SDC-1 expression increases significantly—up to 15-fold—within 24 h post-injury in keratinocytes at the wound edge [39]. However, some studies have reported reduced expression in actively migrating cells, suggesting context-dependent regulation. SDC-1-deficient mice show impaired wound healing marked by decreased keratinocyte activation, proliferation, and migration, notably lacking the early proliferative burst observed in normal healing [40]. Conversely, SDC-1 overexpression enhances neonatal keratinocyte proliferation [41]. In dermabrasion models, SDC-1-deficient mice display delayed cell proliferation and reduced epidermal localization of α9 integrin [40]. Moreover, TGF-β1 induces integrin expression (e.g., α2β1, αvβ6, αvβ8, and α6β4) in wild-type keratinocytes, but not in SDC-1-deficient cells [42], suggesting that SDC-1 modulates integrin expression via TGF-β signaling.

SDC-1 also functions as a co-receptor for laminin-332, promoting keratinocyte migration. Specifically, the G45 domain of laminin-332 induces MMP-9 expression via SDC-1, and the α3 LG45 domain recruits both SDC-1 and CD44 to form keratinocyte podosomes. These podosomes, enriched with MMP-9 and MMP-14, mediate localized ECM degradation around actin cores on gelatin and collagen substrates—facilitating cell migration and re-epithelialization of the wound surface [43,44].

SDC-4 plays a central role in fibroblast invasion by binding to the Hep II domain of fibronectin, promoting migration into the fibrin-rich provisional matrix. This is essential for granulation tissue formation and ECM deposition. PDGF enhances SDC-4 expression at both the mRNA and protein levels [45], further supporting fibroblasts’ function.

In fibroblasts, SDC-1 expression is induced by FGF-2. In 3D collagen matrices, SDC-1 enhances dermatan sulfate synthesis, which promotes FGF-7 activation—facilitating ECM remodeling and fibroblast activity [46,47,48]. Moreover, SDC-1 regulates integrin activation in fibroblasts; SDC-1-deficient fibroblasts show increased αvβ1 integrin expression and exhibit faster migration [42], indicating a compensatory response in the absence of SDC-1.

### 2.4. Regulation of the Remodeling Phase by SDCs

Tissue remodeling is the final and longest phase of wound healing, characterized by type III collagen replacement with type I collagen, ECM reorganization, reduction in cellularity, and vascular regression. Fibroblasts play a key role in the synthesis of collagen and other matrix components, while gradually replacing type III collagen with stronger type I collagen. Proper spatial reorganization of fibroblasts and ECM components is essential to prevent excessive scarring or chronic wounds [14,49]. Recent studies have highlighted that SDCs, particularly SDC-4 and SDC-1, act not only as adhesion molecules but also as dynamic regulators of fibroblast behavior, growth factor signaling, cytoskeletal remodeling, and ECM architecture [13,18,43,50].

SDC-4 is pivotal in organizing the ECM and directing fibroblast functions. It modulates fibroblast adhesion and spatial patterning by regulating EphA2 signaling via PKCα, switching fibroblasts from repulsive to adhesive phenotypes, which aids matrix alignment and contraction [50]. It also integrates TGF-β1/MEK/ERK signaling to promote myofibroblast differentiation and α-SMA-mediated contraction [49]. In vivo, SDC-4 deficiency leads to delayed wound healing and impaired angiogenesis, with reduced vessel density and defective matrix contraction [51]. This is linked to disrupted FAK/RhoA signaling and impaired actin stress fiber formation. Notably, SDC-4’s function is spatially regulated: while it promotes matrix contraction at the wound center, its interaction with tenascin-C at the wound margins modulates fibronectin–SDC-4 binding, thereby preventing excessive contraction [52]. This spatial regulation ensures coordinated ECM remodeling for optimal wound resolution. While essential for repair, excessive SDC-4 activity may contribute to fibrosis via ERK overactivation [49]. Through interactions with CCN2 and fibronectin receptors, SDC-4 facilitates focal adhesion and fibroblast anchorage via ERK/FAK signaling during late-stage remodeling [53]. Moreover, SDC-4 mediates antifibrotic pathways: wound-homing peptides (e.g., CAR peptides) activate SDC-4-dependent ARF6 signaling, enhancing keratinocyte migration and reducing myofibroblast accumulation, suggesting a role in limiting fibrosis [13].

SDC-1 plays a crucial role in the tissue remodeling phase by regulating fibroblast adhesion, migration, and ECM interaction through TGF-β1/PKCα and integrin-mediated signaling pathways. In SDC-1-deficient fibroblasts, enhanced migratory behavior and altered integrin activity suggest a regulatory role for SDC-1 in maintaining appropriate cell–matrix communication [42]. Interestingly, SDC-1 is upregulated in keloid tissues and fibroblasts, where its expression correlates with increased levels of ECM proteins such as α-SMA, fibronectin, pro-collagen I, and collagen III. Knockdown of SDC-1 significantly reduces these ECM components in both cell lysates and culture media, likely via modulation of TGF-β1/Smad and MAPK signaling pathways [54]. These findings indicate that SDC-1 promotes matrix accumulation and may contribute to fibrotic remodeling when overexpressed. Paradoxically, loss of SDC-1 can also enhance TGF-β1 signaling and impair keratinocyte migration, suggesting that SDC-1 may exert context-dependent effects. In this case, SDC-1 deficiency skews remodeling toward fibrosis rather than regeneration by disrupting epithelial–mesenchymal communication [55].

In addition to its membrane-bound form, shed SDC-1 ectodomains exert distinct biological effects. Soluble SDC-1 fragments have been shown to disrupt ECM remodeling, impair keratinocyte migration, and promote abnormal angiogenesis by degrading elastin and inhibiting FGF-2 signaling [18,56]. These observations suggest that excessive SDC-1 shedding may act as a negative regulator of late-stage wound healing. However, in acute wounds, shed ectodomains of SDC-1 and SDC-4 also serve protective roles by binding to neutrophil-derived proteases such as cathepsin G and elastase through their heparan sulfate chains. This interaction protects the enzymes from serpin-mediated inhibition, helping to maintain a controlled proteolytic environment required for effective ECM turnover and growth factor activation [57]. These dual functionalities illustrate the context-dependent roles of SDCs: while excessive or dysregulated shedding may hinder repair, controlled ectodomain release contributes to balanced matrix remodeling and the restoration of normal tissue architecture.

Compared to SDC-1 and SDC-4, SDC-2 is less studied but increasingly linked to fibrotic remodeling. It is overexpressed in keloid fibroblasts, with shedding promoted by epithelial–mesenchymal interactions and elevated FGF-2. Shed SDC-2 may enhance profibrotic signaling and abnormal ECM accumulation [58].

SDCs act as integrative platforms coordinating fibroblast activity, ECM organization, growth factor signaling, and proteolytic balance. While they support normal tissue regeneration, their dysregulation—especially of SDC-1 and SDC-4—can skew healing toward fibrosis, delayed repair, or chronic scarring [51]. These findings underscore the potential of targeting SDCs for therapeutic modulation of wound healing and fibrosis.

## 3. Conclusions

SDCs, particularly SDC-1 and SDC-4, play integral roles in maintaining skin homeostasis and coordinating the complex cellular responses required for effective wound healing. These transmembrane heparan sulfate proteoglycans function not only as structural components but also as dynamic signaling platforms, interacting with a variety of ligands such as growth factors, integrins, and ECM proteins. In healthy skin, SDC-1 and SDC-4 are involved in keratinocyte differentiation and vascular stability [5,53]. Both are expressed in endothelial cells, where they reduce permeability, stabilize the vasculature via interactions with fibrinogen [7,8], and contribute to anticoagulant activity through their heparan sulfate chains [9,10]. However, these functions are often disrupted in the early stages of wound healing, emphasizing the need for rapid and regulated SDC activity during repair.

During the proliferation phase, SDC-1 and SDC-4 coordinate keratinocyte and fibroblast activity. SDC-1 promotes keratinocyte proliferation and migration by acting as a co-receptor for growth factors and supporting ECM remodeling via laminin-332 and MMP induction [43,44]. SDC-4 facilitates fibroblast migration and angiogenesis, partly through its interactions with fibronectin and PDGF signaling [45], and is now recognized as a key regulator of intussusceptive angiogenesis [59] and a critical molecular target in diabetic wound healing, identified through single-cell RNA sequencing and machine learning approaches [60]. Both SDCs regulate integrin activation and cytoskeletal dynamics, processes essential for re-epithelialization and granulation tissue formation. In addition, previous research has demonstrated that SDC-4 works in concert with integrins to regulate epithelial cell migration and focal adhesion dynamics, establishing a conceptual basis for understanding syndecan-mediated re-epithelialization during wound healing [61,62].

In the remodeling phase, SDC-4 governs ECM reorganization and fibroblast contractility through PKCα–EphA2 and TGF-β1–ERK signaling [50], enabling myofibroblast differentiation and controlled matrix contraction. Its spatial regulation—promoting contraction centrally while limiting it at wound margins—prevents fibrosis, though sustained activation may lead to pathological scarring. SDC-1 also contributes by modulating fibroblast adhesion and migration via integrin and PKCα pathways [42]. While its shed ectodomains help balance proteolysis, excessive shedding disrupts keratinocyte migration and angiogenesis. Elevated SDC-1 in fibrotic tissues such as keloids [54] further suggests its role in driving ECM overproduction through TGF-β/Smad and MAPK pathways. Moreover, the remarkable specificity in growth factor binding to SDCs, particularly heparin-binding growth factors [63], underscores their fine-tuned regulation in wound repair and fibrosis.

SDCs are central regulators of wound healing, balancing regeneration and fibrosis by integrating extracellular signals and modulating downstream pathways. Dysregulation of SDC expression or shedding is linked to chronic wounds and fibrotic diseases, highlighting their therapeutic potential. Targeting SDC-related pathways—such as controlling SDC-4 signaling or limiting SDC-1 shedding—may enhance tissue regeneration while preventing excessive scarring. Given the shared mechanisms across epithelial tissues, insights derived from skin repair may also apply to other organs. Future research should focus on dissecting the context-dependent functions of individual SDC isoforms using single-cell multi-omics, spatial transcriptomics, and advanced imaging approaches. Integrating these data with computational modeling and machine learning could reveal predictive markers of healing outcomes and identify new therapeutic targets. Moreover, understanding how SDCs interact with other signaling networks (e.g., integrin, growth factor, and mechanotransduction pathways) in a time- and location-specific manner will be crucial for designing precision therapies that promote regeneration without fibrosis.

In conclusion, SDC-1 and SDC-4 perform distinct but complementary functions throughout all stages of wound healing, from epithelial regeneration to angiogenesis and matrix remodeling. Deepening our understanding of their molecular mechanisms and regulatory networks will pave the way for innovative regenerative and antifibrotic therapies across diverse tissue types.

**Table 1 ijms-26-10571-t001:** Cell-specific syndecan expression and functions in each phase of skin wound healing.

Phase	Key SDCs	Alterations of SDCs	Syndecans’ Functions	References
Hemostasis	SDC-1 SDC-4	Cell surface clustering	- SDC-1 reinforces endothelial barrier integrity by binding fibrinogen.	[8]
			- SDC-4 enhances platelet aggregation by clustering.	[19,20]
		Elevated shedding of SDCs	- Shed SDC-1/SDC-4 competitively inhibit growth factor signaling by sequestering ligands from their receptors.	[15]
			- Shed SDC-1/-4 bind and modulate proteases, altering proteolytic balance and potentially affecting clot stability and remodeling.	[15,16]
			- Excessive shedding of SDC-1 promotes leukocyte adhesion and increases the risk of thrombosis by glycocalyx loss-induced cell exposure and reduced thrombin generation.	[21,22]
[Inflammation	SDC-1 SDC-4	Upregulated SDC levels	- SDC-1 suppresses early inflammation by delaying leukocyte infiltration, edema, and inflammatory mediator expression.	[23]
			- SDC-1 enhances neutrophil–endothelial interaction and transmigration by binding IL-8.	[25,26,27]
			- SDC-4 is strongly induced in the epidermis after skin injury.	[51]
			- SDC-1 is initially upregulated in endothelial cells and keratinocytes.	[39]
			- SDC-4 promotes eosinophil motility via AT-III through PKC, PI3K, and PDE signaling, while inhibiting chemokine-driven migration.	[24]
		Elevated shedding of SDCs	- SDC-1 shedding disrupts the glycocalyx and increases VEGF secretion by keratinocytes.	[29]
			- Shed SDC-1 reduces neutrophil migration by sequestering IL-8 and disrupting its gradient.	[15,28]
Proliferation	SDC-1 SDC-4	Upregulated SDC levels in keratinocytes	- SDC-1 promotes keratinocyte proliferation via growth factor signaling.	[40,41]
			- SDC-1 enhances keratinocyte migration by upregulating integrins via TGF-β signaling.	[42]
			- SDC-1 induces laminin-332-mediated MMP-9 expression, promoting ECM degradation, podosome formation, and re-epithelialization.	[43,44]
			- SDC-4 promotes keratinocyte migration via epidermis-selective expression.	[13]
		Upregulated SDC levels in fibroblasts	- SDC-1 enhances dermatan sulfate synthesis, activating FGF-7 and promoting ECM remodeling and fibroblast activity.	[46,47,48]
			- SDC-1 regulates fibroblast adhesion and migration via integrin expression.	[42]
			- SDC-4 promotes fibroblast migration by binding fibronectin.	[45]
Remodeling	SDC-1 SDC-4 SDC-2	Upregulated SDC levels in fibroblasts	- SDC-1 inhibits fibroblast migration and promotes adhesion via TGF-β1–PKCα signaling.	[42,55]
			- Excessive SDC-4 activity drives fibrosis via ERK overactivation.	[53]
			- SDC-4 switches fibroblasts to an adhesive phenotype, aiding matrix alignment and contraction via EphA2–PKCα signaling.	[50]
			- SDC-4 facilitates fibroblast focal adhesion via CCN2/fibronectin–ERK/FAK signaling.	[53]
			- SDC-4 promotes myofibroblast differentiation via TGF-β1/MEK/ERK signaling.	[49]
			- SDC-4 promotes angiogenesis by regulating actin stress fiber formation via FAK/RhoA signaling.	[51,52]
			- SDC-4 mediates antifibrotic pathways by reducing myofibroblast accumulation.	[13]
			- Excessive SDC-4 activity contributes to fibrosis via ERK overactivation.	[49]
			- SDC-1 is induced in keloid tissues/fibroblasts, driving ECM accumulation via TGFβ1/Smad and MAPK.	[54]
			- SDC-2 is overexpressed in keloid fibroblasts.	[58]
		Elevated shedding of SDCs	- Shed SDC-2 may enhance profibrotic signaling and abnormal ECM accumulation.	[58]
			- In acute wounds, shed SDC-1/SDC-4 preserve protease activity, supporting ECM turnover and growth factor activation.	[57]
			- Shed SDC-1 impairs ECM remodeling, keratinocyte migration, and promotes abnormal angiogenesis by degrading elastin and inhibiting FGF-2 signaling in late wound healing.	[18,56]

## Figures and Tables

**Figure 1 ijms-26-10571-f001:**
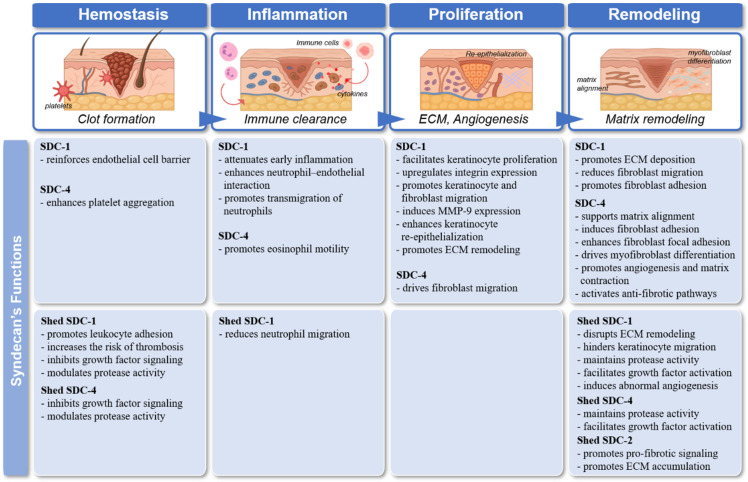
The stages of skin wound healing and syndecans’ functions.

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
