# Peer review of "Spatial and Functional Roles of Syndecans in Skin Wound Healing"

_ijms, 2025, doi:10.3390/ijms262110571_

Round 1
Reviewer 1 Report
Comments and Suggestions for Authors
This is a wonderful, comprehensive review of syndecans and I congratulate the authors on crafting this nice manuscript. A few comments:
-Line 126 has syndecan-1 abbreviated as Sdc1 while it is otherwise abbreviated as SDC-1. This should be consistent throughout the manuscript.
-The only other potential improvement that might be made in this well-written manuscript is to expand upon future potential research directions. There is a short mention of this in the conclusions, but calling attention to various potential research questions throughout the body of the paper could add value and highlight what remains to be learned about these important proteoglycans.
Author Response
This is a wonderful, comprehensive review of syndecans and I congratulate the authors on crafting this nice manuscript. A few comments:
(1) Line 126 has syndecan-1 abbreviated as Sdc1 while it is otherwise abbreviated as SDC-1. This should be consistent throughout the manuscript.
Response: We appreciate the reviewer for noting this inconsistency. The abbreviation for syndecan-1 has been standardized as “SDC-1” throughout the entire manuscript.
(2) The only other potential improvement that might be made in this well-written manuscript is to expand upon future potential research directions. There is a short mention of this in the conclusions, but calling attention to various potential research questions throughout the body of the paper could add value and highlight what remains to be learned about these important proteoglycans.
Response: Thank you for this valuable suggestion. We have expanded the “Conclusions” section to include a more detailed discussion on emerging research directions. Specifically, we now emphasize: (1) the potential of multi-omics and single-cell approaches to elucidate cell-type–specific functions of SDCs and (2) the therapeutic targeting of SDC-mediated signaling pathways in tissue regeneration and fibrosis. These additions also reflect the insightful comments provided by Reviewer 2, ensuring a more comprehensive outlook on future research perspectives.
Reviewer 2 Report
Comments and Suggestions for Authors
The article is well-written and addresses most of the topics related to syndecans in wound healing. The review would benefit from the addition of few recent articles that would add further insight into the review as well as presenting a deeper understanding of the syndecan biology than is presented at the moment (please see below).
- single-cell RNA sequencing, transcriptomic analysis and machine learning identifies SDC4 as the most appealing molecular target in diabetic wound healing. Please cite (PMID: 39557273). This article highlights the importance of syndecans for wound healing and tissue regeneration in general.
- SDC4 is vital for intussusceptive angiogenesis, please cite (PMID: 33596666)
- Please review the function/biology of SDC4 according to the recent review article by Dr. Couchman (https://doi.org/10.1002/pgr2.70018).
- The remarkable specificity in growth factor binding to syndecans has been demonstrated, please see (PMID: 30952866). As syndecans are co-receptors for some of the most important growth factors for wound healing, the authors need to review the heparin-binding growth factor binding to syndecans. Please see the classic study by Simons demonstrating the remarkable specificity in hepartin-binding growth factor binding to syndecans (PMID: 30952866)
- The classic work by the Humphries-lab demonstrating syndecan-mediated epithelial migration is missing. The pathway´s translation (ref # 13) is beautifully illustrated in the manuscript, but the cell migration pathway should be described. Please cite the original studies (PMID: 21982645 and PMID: 23453597).
- The epidermis-selective expression of syndecan 4 in skin wound is demonstrated in ref. # 13, please add to the Table.
The English presentation can be further improved.
Author Response
The article is well-written and addresses most of the topics related to syndecans in wound healing. The review would benefit from the addition of few recent articles that would add further insight into the review as well as presenting a deeper understanding of the syndecan biology than is presented at the moments (please see below).
(1) single-cell RNA sequencing, transcriptomic analysis and machine learning identifies SDC4 as the most appealing molecular target in diabetic wound healing. Please cite (PMID: 39557273). This article highlights the importance of syndecans for wound healing and tissue regeneration in general.
(2) SDC4 is vital for intussusceptive angiogenesis, please cite (PMID: 33596666).
(3) Please review the function/biology of SDC4 according to the recent review article by Dr. Couchman (https://doi.org/10.1002/pgr2.70018).
Response: Thank you for this valuable suggestion. We have expanded the “Conclusions” section to include a more detailed discussion on emerging research directions, along with the specific citations as recommended by the reviewer. The revised portion of the Conclusion now reads as follows: “SDC-4 facilitates fibroblast migration and angiogenesis, partly through its interactions with fibronectin and PDGF signaling [45], and is now recognized as a key regulator of intussusceptive angiogenesis [61] and a critical molecular target in diabetic wound healing identified by single-cell RNA sequencing and machine learning approaches [62]. Both SDCs contribute to integrin activation and cytoskeletal reorganization, processes essential for re-epithelialization and granulation tissue formation.”
(4) The remarkable specificity in growth factor binding to syndecans has been demonstrated, please see (PMID: 30952866). As syndecans are co-receptors for some of the most important growth factors for wound healing, the authors need to review the heparin-binding growth factor binding to syndecans. Please see the classic study by Simons demonstrating the remarkable specificity in hepartin-binding growth factor binding to syndecans (PMID: 30952866)
Response: Thank you for this valuable suggestion. We have incorporated this point into the “Conclusions” section as follows: “Moreover, the remarkable specificity in growth factor binding to SDCs, particularly heparin-binding growth factors [65], underscores their fine-tuned regulation in wound repair and fibrosis.”
(5) The classic work by the Humphries-lab demonstrating syndecan-mediated epithelial migration is missing. The pathway´s translation (ref # 13) is beautifully illustrated in the manuscript, but the cell migration pathway should be described. Please cite the original studies (PMID: 21982645 and PMID: 23453597).
Response: We added a brief description of syndecan-4–mediated epithelial migration and cited the original studies (PMID: 21982645; 23453597). The following sentence was inserted in the “Conclusion” section: “Moreover, SDC-4 has been shown to act in concert with integrins to regulate epithelial cell migration and focal adhesion dynamics, providing a conceptual framework for understanding SDC-mediated re-epithelialization during wound healing [63,64].”
(6) The epidermis-selective expression of syndecan 4 in skin wound is demonstrated in ref. # 13, please add to the Table.
Response: We have revised Table 1 to include the epidermis-selective expression of SDC-4 in skin wounds as demonstrated in Ref. #13.